# End-to-End Scene Augmentation for Robust Robot Manipulation Learning

**Chengbo Yuan**[*]
student ID: 2024211275

**Shaoting Zhu**[*]
Student ID: 2024311565

**Suraj Joshi**[*]
Student ID: 2024280205

## 1 Background

We have seen huge progress in developing foundation models in the domain of vision and language. The foundation models are very generalizable and can be used for multiple tasks with zero-shot or few-shot learning. However, we have not seen good foundation models for robotics which can be attributed to the lack of large-scale diverse datasets for robotics. Collecting large-scale datasets with large variations in the scenes requires engineering heavy automation or laborious teleoperations using humans. Some methods aim to solve this issue by generating datasets using simulated, but simulated datasets fail to capture the underlying distribution of the real world and there also exists some gap in sim2real transfer.

In this work, we aim to use diffusion models to augment new samples into an existing dataset by manipulating background scenes so that learned policies using the augmented dataset can be applied to a broad range of robot learning tasks. We finally aim to train a policy network using the augmented dataset, which can be utilized in diverse downstream tasks.

## 2 Related works

**Visual Data Augmentation for robot learning.** Some of the early techniques focus on studying how different perturbations on the visual dataset e.g. changing lighting, scaling, and much more can lead to better generalization[1, 2]. However, these techniques can only robustly handle the same scene under varied conditions. Some of the more modern techniques use diffusion models to diversify the scenes by adding new objects and distractors to the scenes in the existing dataset[3, 4]. They are also not very scalable as they are not completely generating new scenes instead manipulating the existing scenes by adding some objects. Also, they require textual input to modify the scene in the given image, making these processes time-consuming. GreenAug[5] tries to completely generate new scenes from the existing scene using chroma key algorithms. Still, it requires data to be collected with green backgrounds and the scenes overlayed are some predefined scenes that might not be semantically compatible with the object to be manipulated. We aim to generate completely new scenes that capture the underlying physics of the world and are also semantically compatible with the object to be manipulated in the scene.

## 3 Proposed Method

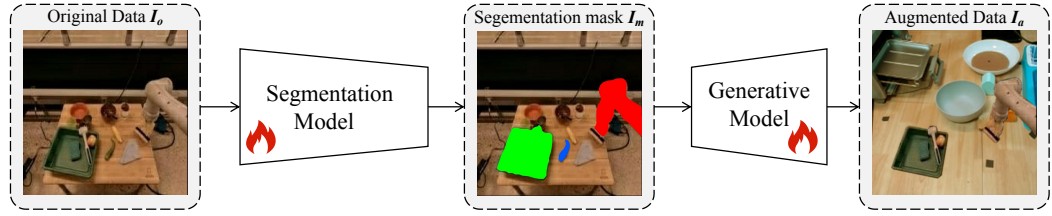

Figure 1: Overview pipeline of our proposed method. We train a segmentation model and a generative model, and augment the robot dataset by changing the image's background.

### 3.1 Math Definition

Given an unseen robot manipulation dataset $\mathcal{D}$, our goal is augmenting the $\mathcal{D}$ to $\mathcal{D}_a$ using a trained segmentation model and a generative model. We change the background while making the robot arm and the object part the same. For downstream tasks, we may use this augmented dataset to train a generalizable robust robot action policy, $\pi(\mathcal{D}_a)$.

---

[*]Equal contribution (authors listed in alphabetical order).

Preprint. Course project.

### 3.2 Basic Pipeline

Specifically, we handle this task in following three steps.

**1)** We first create a dataset of <RGB image, segmentation mask> pair. We use segmentation models to obtain masks for real-world datasets, while for simulated datasets, masks can be obtained from the simulator.

**2)** We then fine-tune a segmentation model and a generative model, especially for the robotic arm images. The segmentation model takes in an RGB image and outputs a segmentation mask. The generative model takes in the segmentation mask and outputs an RGB image. In addition, we can use framework in CycleGAN[6] to improve the consistency between original dataset and augmented dataset.

**3)** Finally, we augment an unseen robot manipulation dataset to expand the background. We first use segmentation model to get the mask of object and robot arm $\mathcal{I}_m(\mathcal{M}_i)$, and invert it to get the mask of the background. Then generative model is used to generate images $\mathcal{I}_o$ containing different types of background. The generated images and the original images form the augmented dataset $\mathcal{D}_a$ together.

## 4 Resource Survey

In this section, we provide a brief introduction to the datasets, segmentation labeling, and diffusion model resources that will be used in our project.

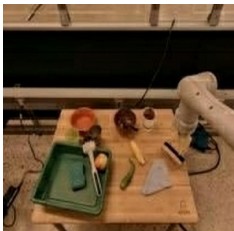 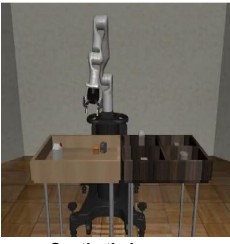 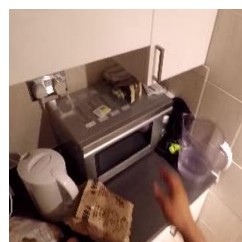

**Real-Robot Images**
(Open-X-Embodiment)

**Synthetic Images**
(RLBench & RoboSuite)

**Human Images**
(Epic-Kitchen)

Figure 2: Three types of data resource we used in our project.

### 4.1 Dataset Resources

Due to the scarcity of real-world robot datasets, we plan to utilize a combination of real-world robot datasets, synthetic datasets, and egocentric human video datasets to train our model (Figure 2).

**Real-world Robot Images (Open-X-Embodiment [7]).** For real-world robot images, we will use the Open-X-Embodiment (OXE, [7]) dataset, which is the largest collection of real-world robot data that over 20 robot datasets. For each task in OXE, we randomly select one trajectory and sample 8 images from it, resulting in a total of 4,310 third-person view images.

**Sythetic Robot Images (RLBench[8] and RoboSuite[9]).** Generating synthetic images from simulation is more convenient, though such images often lack realistic textures and physics. We aim to generate approximately 1,500 simulated images.

**Egocentric Human Images (Epic-Kitchens [10]).** Egocentric human videos provide valuable insights into hand-object interactions in the physical world, which share similarities with robot manipulation. As a supplementary data source, we will use the Epic-Kitchens dataset [10], extracting around 4,000 images to enhance model training.

### 4.2 Segmentation Labeling

For real-robot images, we find that the segmentation performance of off-the-shelf models like GroundingDINO [11] and the Segment Anything Model (SAM) [12] is suboptimal. So we decided to train our own segmentation model. We will manually annotate the real-robot images for segmentation model training, leveraging the open-source ISAT tool [13] in combination with SAM [12] to improve labeling efficiency. For synthetic data, we can directly obtain robot and object masks from simulation. For human videos, we will use the state-of-the-art hand-object segmentation model EgoHOS [14] to generate masks for hands and manipulation objects.

### 4.3 Diffusion Model

Once segmentation for the robot and manipulated objects is complete, the diffusion model will be used to generate entire images, with background inpainting. To achieve this, we plan to train ControlNet [15], one of the most widely used models for conditional image generation. Additionally, more advanced models such as Uni-ControlNet [16] may also be considered.

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
