# OpenReview forum: "[Proposal-ML] End-to-End Scene Augmentation for Robust Robot Manipulation Learning"
_tsinghua.edu.cn/THU/2024/Fall/AML — THU 2024 Fall AML Submission_

### Official Review · ~Zhen_Leng_Thai1 · 2024-11-08
**Great and Detailed Proposal with Outstanding Ideas**

**Rating:** 10
**Confidence:** 4

**Review:**

This paper introduces an innovative method using diffusion models to augment datasets by manipulating background scenes. This method includes a segmentation model for mask segmentation and generative model for image generation. The background and mathematical definitions are well-defined. Relation works about segmentation, diffusion models and visual data augmentation is comprehensive. The proposed method is structured in three steps: dataset creation, model fine-tuning, and model integration, with additional details on dataset resources, segmentation labeling, and diffusion model specifics.

---

### Official Review · ~Xiying_Huang2 · 2024-11-09
**Innovative Scene Augmentation for Enhanced Robot Manipulation Learning**

**Rating:** 10
**Confidence:** 4

**Review:**

This paper proposes a method for enhancing robot manipulation learning through scene augmentation. The approach focuses on using diffusion models to expand existing datasets by altering background scenes, thereby improving policy networks for diverse robotic tasks.

Quality: The methodology is clear and methodical, incorporating segmentation and generative models. Each stage of the process is detailed and feasible.

Clarity: The paper is generally well-structured. However, technical aspects, such as specific model training details, could be expanded for further clarity.

Originality: The proposal is innovative in its use of diffusion models for generating diverse robotic training environments, contributing novel insights into dataset augmentation for sim-to-real transfer.

Significance: Given the challenges of sim-to-real transfer in robotics, this research is relevant and could have substantial impacts on the field.

Pros:
	•	Addresses a key challenge in robotic manipulation learning.
	•	Proposes a feasible method with clear steps.
	•	Utilizes a variety of data sources to ensure robust model training.

Cons:
	•	Limited discussion on potential limitations of the diffusion models.
	•	Further clarification on quantitative evaluation metrics would be beneficial.

---

### Official Review · ~Yufei_Zhuang1 · 2024-11-09
**Great potential and well formed**

**Rating:** 9
**Confidence:** 4

**Review:**

This project shows great potential in the field of robot learning. By focusing on dataset augmentation with diffusion models, it tackles a significant issue - the scarcity of large - scale and diverse datasets in robotics.
 The approach of background scene manipulation is clever. It's an innovative way to generate new samples within existing datasets, which can be crucial for improving the generalization of learned policies. This has far - reaching implications for various robot learning tasks as it can expose the models to more diverse scenarios.
Using a combination of real - world, synthetic, and human - centric datasets is also a strong point. This diverse data source can enrich the training process and potentially lead to more robust models. Overall, this project seems to be on a promising path to enhance robot learning capabilities.

---

### Official Review · ~Junjie_Chen8 · 2024-11-11
**Good Proposal**

**Rating:** 8
**Confidence:** 4

**Review:**

The proposal provides a clear and well-structured approach to addressing a significant challenge in robotics: augmenting datasets to improve manipulation learning. The use of diffusion models for scene augmentation and the combination of real-world, synthetic, and egocentric human datasets make the methodology both innovative and practical. The pipeline is systematically designed, incorporating segmentation and generative models to enhance dataset diversity and robustness. What's more, I think a clearer description of evaluation metrics would strengthen the proposal.

---

### Official Review · ~Chengming_Shi1 · 2024-11-11

**Rating:** 10
**Confidence:** 3

**Review:**

### Summary

The proposal “End-to-End Scene Augmentation for Robust Robot Manipulation Learning” aims to address the lack of diverse datasets in robotics by using diffusion models to augment existing datasets with varied background scenes. The goal is to enhance the generalizability of robot manipulation policies across a wide range of tasks by training on an augmented dataset that reflects a broader set of environmental conditions.

### Pros

1. **Innovation in Data Augmentation**: The use of diffusion models for scene augmentation is a novel approach that could significantly improve the diversity and robustness of robot learning datasets.
2. **Potential for Generalization**: By augmenting the dataset with a variety of backgrounds, the trained policies are more likely to generalize to new environments, reducing the sim-to-real gap.
3. **Integration of Real and Synthetic Data**: The proposal’s strategy of combining real-world robot images, synthetic robot images, and egocentric human images could lead to a rich and varied training dataset.
4. **Custom Segmentation Model**: Training a custom segmentation model for robot and object masks indicates a commitment to high-quality data preprocessing, which is crucial for the success of the generative model.
5. **Advanced Generative Models**: The use of ControlNet and Uni-ControlNet for conditional image generation suggests that the proposal is leveraging state-of-the-art technology to achieve its goals.

### Cons

1. **Complexity of Implementation**: The proposed method involves multiple steps, including dataset creation, model training, and augmentation, which could be complex and time-consuming to implement.
2. **Dependency on High-Quality Masks**: The success of the generative model heavily relies on the accuracy of the segmentation masks, which may be challenging to obtain, especially for real-world data.

---

### Official Review · ~Zhang_Mingkang1 · 2024-11-11
**Novel idea**

**Rating:** 9
**Confidence:** 3

**Review:**

Strengths:

Background


Excellent identification of the data scarcity problem in robotics.
Clear connection to foundation models and their limitations.
Strong motivation for bridging sim-to-real gap.
Well-articulated research challenge.


Definition


Precise definition of the transformation from D to Da.
Well-defined policy learning objective π(Da).


Related Work


Comprehensive review of visual data augmentation techniques.
Excellent critique of existing methods' limitations.
Strong analysis of current approaches (GreenAug, diffusion models).
Clear identification of research gaps.


Proposed Method


Innovative three-step pipeline.
Well-thought-out dataset combination strategy.
Clear technical approach using state-of-the-art tools.
Detailed resource survey and implementation plan.

Key Comments:

Outstanding integration of multiple data sources (real-robot, synthetic, human).
Novel use of diffusion models for background generation.
Practical approach to segmentation using modern tools.
Well-structured experimental design.
Excellent consideration of data quality and diversity.

Areas for Improvement:

Might benefit from quantitative metrics for evaluating generated scenes.
Consider discussing computational requirements for training.
Could elaborate on potential failure modes.

This is an exceptionally well-crafted proposal that addresses a fundamental challenge in robot learning. The approach is innovative, practical, and well-grounded in current literature.

---

### Official Review · ~Jiuyang_Zhou1 · 2024-11-12
**creative and sufficient work**

**Rating:** 9
**Confidence:** 4

**Review:**

This paper proposes using diffusion models to augment existing datasets by manipulating background scenes, aiming to address the issue of the lack of large-scale and diverse datasets in the field of robotics. Its main contribution lies in presenting a new scene augmentation method, which is specifically implemented through a segmentation model and a generative model. First, a dataset of <RGB image, segmentation mask> pairs is created. Then, the models are fine-tuned and the CycleGAN framework is utilized to ensure consistency. Finally, the dataset is augmented. The paper employs a variety of resources, including datasets such as Open - X - Embodiment, segmentation labeling methods like manual annotation, and diffusion models such as ControlNet. However, this method may face some challenges, such as the effectiveness and training difficulty of the generative model, and the compatibility of the new scenes with the real world. Overall, it provides an innovative data augmentation idea for the field of robot manipulation learning and is expected to enhance the generalization ability and robustness of the policy network in downstream tasks.

---

### Official Review · ~Chaoqun_Yang2 · 2024-11-12
**A meaningful proposal**

**Rating:** 8
**Confidence:** 4

**Review:**

**Summary:**
The paper tackles the critical issue of dataset diversity in robotics, which is essential for developing generalized robotic manipulation models. The problem is rooted in practical applications, particularly in the deployment of robots in varied and dynamic real-world environments. Solving this problem could significantly impact the field by enabling robots to perform tasks across different settings more effectively. The proposed method involves using diffusion models to augment datasets by manipulating background scenes, aiming to train a more generalized policy network for robotic manipulation tasks.

**Highlights:**
1. **Diffusion Models for Scene Augmentation:** The use of diffusion models for scene augmentation is innovative and represents a significant departure from traditional data augmentation techniques. This approach could potentially bridge the gap between simulation and real-world applications, a notorious challenge in robotics.
2. **Comprehensive Dataset Strategy:** The plan to utilize a combination of real-world, synthetic, and egocentric human video datasets is commendable. This strategy could result in a more robust and versatile model capable of handling various manipulation tasks.

**Advice:**
1. **Detailed Comparison with Related Work:** While the paper mentions related works, a more detailed analysis of existing approaches, their limitations, and how the proposed method overcomes these limitations is necessary. This should include a discussion on the sim2real transfer problem and how the proposed method addresses it.
2. **Experimental Design and Baseline Comparisons:** The paper would be better if presenting  more detailed experimental design, including the the baseline in this area, the metrics for evaluating the method's performance, and some sub-experiments to explore the effectiveness of different modules . It is crucial to describe how the proposed method will be implemented and validated against these baselines.

---

### Official Review · ~Kairong_Luo1 · 2024-11-12
**Interesting dataset generation**

**Rating:** 8
**Confidence:** 4

**Review:**

Strength:
1. Complete preparation: dataset basics, and so on;
2. A interesting combination of diffusion model and segmentation models;

Weakness:
1. Questions occur to me that why does diffusion model bypass the sim-real gap shown in other synthetic data?
2. What is the advantages about fixing the segmentation masks than other options? Because it seems fix the position of objects in the scenes?

---

### Official Review · ~Justinas_Jučas3 · 2024-11-12
**Unique Idea and Solid Proposal**

**Rating:** 8
**Confidence:** 4

**Review:**

In general, the proposal is ambitious, and for sure original. It is also very well structured, with added visualizations, which make it easier to view.

## Advantages
1. Well structured and easy to read. All requirements satisfied.
2. Original idea
3. Without a doubt, achievable within the time constraints
4. Detailed pipeline explanaition: already providing a clear idea of how the algorithm would work

## Disadvantages
It is not really mentioned how the influence of your augmented data will be evaluated. How will you know that your technique gives positive results for actual robot training? Perhaps this is not within the scope of the proposal, but anyone can slightly augment any image, but what matters is how it contributes to the end goal (in our case it is better-performing robots? or perhaps something else, maybe you only want to diversify the dataset).

Without clear evaluation metrics, how will you make sure that the way you augment the dataset does not, for instance, disbalance it, reducing the actual important features it had?

---

### Official Review · ~Zihan_Wang7 · 2024-11-12
**mocking dataset generation task**

**Rating:** 9
**Confidence:** 4

**Review:**

**Strengths:**
* Clear technical pipeline with practical implementation steps
* Good use of modern tools (ControlNet, SAM) and diverse data sources
* Well-structured methodology for maintaining object/robot consistency

**Concerns:**
1. No discussion of how to ensure physical plausibility in generated scenes
2. Missing analysis of potential impact on policy learning - could complex backgrounds hurt performance?
3. Unclear evaluation framework for measuring generalization benefits

**Recommendations:**
1. Include methods to verify physical consistency in generated scenes
2. Develop clear validation approach for testing policy network